# SlateFree: a Model-Free Decomposition for Reinforcement Learning with Slate Actions

## Abstract

We consider the problem of sequential recommendations, where at each step an agent proposes some slate of $N$ distinct items to a user from a much larger catalog of size $K >> N$. The user has unknown preferences towards the recommendations and the agent takes sequential actions that optimise (in our case minimise) some user-related cost, with the help of Reinforcement Learning. The possible item combinations for a slate is $\binom{K}{N}$, an enormous number rendering value iteration methods intractable. We prove that the slate-MDP can actually be decomposed using just $K$ item-related $Q$ functions per state, which describe the problem in a more compact and efficient way. Based on this, we propose a novel model-free SARSA and Q-learning algorithm that performs $N$ parallel iterations per step, without any prior user knowledge. We call this method `SlateFree`, i.e. free-of-slates, and we show numerically that it converges very fast to the exact optimum for arbitrary user profiles, and that it outperforms alternatives from the literature.

## 1 Introduction

In many real-life applications, an agent needs to optimally adapt to a random environment through the choice of multi-dimensional actions over time. A specific, common scenario is that of a personalised recommender system (RS), which should pick a set of $N \geq 1$ items among a much larger corpus of size $K >> N$, given the user's recent and past viewing history. An illustrative example of this challenge to solve the top-$N$ recommendations problem is Google's YouTube algorithm, where the current corpus contains several tens of billions of videos Goodrow [2021] and the number of recommended items per view may vary (based on scrolling) but in general involves $N > 20$ items. In such applications the group of $N$ items recommended per step is often called a *slate*.

As Shani et al. [2005] observed, the RS problem is essentially of sequential nature. The recommendation of a specific slate at some point in time offers not only immediate gains if some recommended item is clicked, but can also generate future benefits by guiding the user towards a path of more interesting items as the user session evolves. Shani et al. [2005] formulated this problem within the framework of Markov Decision Processes (MDPs) (Puterman [1994]), and tried to solve it under strong assumptions of independence. Naturally, when the RS needs to learn unknown user preferences, it can do so by observing user-item interactions over time, and the tools of Reinforcement Learning (RL) are most appropriate, see Taghipour et al. [2007] for an early effort.

The optimal slate selection problem per step is in fact combinatorial: the user's choice is affected by the combination (and possibly the order) of the items in the slate, not just their individual importance to the user Aouali et al. [2021]. In such control problems over long horizon, the challenge with slate actions is that the corresponding combinatorial action space is immense and the search for an optimal solution quickly becomes intractable. Even for a small catalog of size $K = 100$ items and a

recommendation slate of size $N = 4$, there are $\binom{100}{4} \approx 4$ million unordered slates as possible actions. Consequently, the number of slate actions in the YouTube example is astronomical.

To tackle the dimensionality explosion in such value iteration algorithms, Ie et al. [2019] (motivated by previous work from Sunehag et al. [2015]) have shown that the slate-value function can be exactly decomposed into $K$ individual $Q$ item-values, per state. Such decomposition could actually render temporal-difference (TD) learning with slates tractable. However, this proposal is based on *prior* knowledge about the user behaviour given any slate and single item choice. In essence this is not a model-free algorithm. It can be implemented if either a user model is assumed, or if the user preference choice per slate is learned from history, which doubles the learning effort of RL and needs to keep in memory $N\binom{K}{N}$ unknowns per user, one unknown per slate and per item choice.

**Our contribution.** We introduce in this work a novel exact decomposition of the Q values for slates, into $K$ individual item-Q values and propose a tractable TD-learning (SARSA- and Q-) algorithm, named here `SlateFree`, which allows to solve efficiently learning and control problems of arbitrary slate dimension $N > 1$ using value-iteration. The important difference compared to Ie et al. [2019] is that our decomposition is entirely model-free in the sense that it does not require any prior knowledge over the user behaviour, and it allows to include costs that depend on both state and action. The proposed decomposition without assuming any independence, simplifies Q-learning considerably:
(i) It keeps $K$ state-item $Q$ functions per state in memory, instead of $\binom{K}{N}$, a massive reduction.
(ii) The optimal slate in the exploitation consists of the $N$-out-of-$K$ items with best item-Q function.

The novel RL algorithm is based on definitions of state-item $Q$ functions, item-costs and transitions. It performs *N-parallel Q-updates* per step, one per item included in the slate. This way, the relevance of an individual item is updated every time this is included in a slate. The method reminds of the independent learners in multi-agent systems, by Claus and Boutilier [1998], one agent per dimension.

The `SlateFree` has considerable performance features. It can learn and take optimal actions over time for any unknown user behaviour. It is shown to be insensitive to the slate-size $N$, thus allowing to scale for arbitrary dimensions. The MDP decomposition is proved in this paper to hold under certain assumptions: (i) the slates are unordered sets of distinct items, and (ii) the user behaviour is Markovian, i.e. the user choice is based only on the current state and recommended slate. Numerical evaluations of the novel RL algorithm show empirically that it finds the optimum even when the cost is a function of both the state and the chosen action-slate, something not possible in other works.

In Section 2 we introduce state-item values as marginal quantities and prove the decomposition of the Bellman equations in the MDP setting. In Section 3 we present the decomposed SARSA- and RL-algorithms, referred to from now on, jointly, as `SlateFree`. In Section 4 we show numerically the exactness of the solution compared to vanilla-RL for users with various preference behaviour and illustrate significant performance improvements against SlateQ in Ie et al. [2019]. We also illustrate how `SlateFree` converges for any type of user and the convergence speed does not depend on the size of the slate. We further illustrate how the algorithm behaves in situations where the cost is a function of both state and action-slate. The code is available on Google Colab SlateFree Authors [2022a] and here SlateFree Authors [2022b].

**Related literature.** The established solution for static recommender systems is based on collaborative filtering, as in Deshpande and Karypis [2004] or matrix factorisation, as in Takács et al. [2008]. Since the problem is actually dynamic, Reinforcement Learning (Sutton and Barto [2018]) is at the moment widely applied to propose more effective or more diverse recommender systems (Karatzoglou et al. [2013], Rohde et al. [2018], Zhou et al. [2020], Warlop et al. [2018]). To overcome the curse of dimensionality in the action space a deep reinforcement learning approach is taken: Zheng et al. [2018] work with a value-based approach and approximate the Q-value by a neural network, whereas Liu et al. [2020b], Liu et al. [2020a] use an actor-critic architecture for policy-based opimisation, where the actor network outputs a continuous feature vector, which can be mapped to an item, thus avoiding the discrete formulation.

RL problems with continuous and high-dimensional action spaces have been recently approached by policy iteration methods. Deterministic policy gradient by (Silver et al. [2014]) is shown to considerably outperform standard policy updates. This method was combined with an efficient mapping to discrete actions by Dulac-Arnold et al. [2015], so that problems like the search for top-$N$ recommendations can be efficiently resolved. Chen et al. [2019] adapt the REINFORCE algorithm with reward independence assumptions. de Wiele et al. [2020] work with amortised inference to

maximise over a smaller subset of possible actions. Metz et al. [2017] propose an autoregressive network architecture to sequentially predict the action value for each action dimension, which requires manual ordering of the actions. Tavakoli et al. [2018] propose a neural architecture with many network branches, one for each action dimension. The special structure of slate-recommendations has given rise to problem specific solutions, like the one by Sunehag et al. [2015], who introduce a formulation that benefits from the fact that at each step the user chooses a single item, for a given action slate. Their approach cannot scale because it needs to keep in memory one value-function per slate. In a very interesting recent approach Ie et al. [2019] show how the slate-value function can be exactly decomposed into individual $Q$ item-values and introduce the method SlateQ. They construct optimal slates from the individual item-values by solving a Linear Program (LP) per step. As mentioned, their decomposition is based on prior knowledge of user choice behaviour.

## 2 Decomposition of slate-MDPs

We first introduce the slate-MDP, defined as $(\mathcal{S}, \mathcal{A}, \mathcal{P}, \mathcal{C}, \lambda)$ and describe the process for the special application of the recommender system. Time is slotted with current step $t$. The state $S_t = s$ at time $t$ will be here the currently viewed item, so the state-space $\mathcal{S} = \mathcal{K}$ is the full item catalog of size $K$. But we can use more general states, e.g. the history of the last $m$-viewed items $m > 1$, so $\mathcal{S} \neq \mathcal{K}$. The action $A_t = \omega$ is an $N$-sized *unordered* slate of recommended items. The set of possible actions $\mathcal{A}$ is the set of all possible unordered $N$-sized slates, where in each slate $\omega \in \mathcal{A}$ no item is duplicated. Here, "unordered" means that only the set of recommended items in the slate is important, not their order. The state transition function $\mathcal{P} : \mathcal{S} \times \mathcal{A} \times \mathcal{S} \rightarrow [0, 1]$ is the probability, given the current state $s$ and recommended slate $\omega$, that the user moves to state $s'$, by either picking one of the recommended items, or rejecting them and selecting some item from the search bar. The general cost function is $\mathcal{C} : \mathcal{S} \times \mathcal{A} \rightarrow \mathbb{R}$ and $\lambda \in (0, 1)$ is the discount rate. The objective is to find an optimal policy $\pi : \mathcal{S} \times \mathcal{A} \rightarrow [0, 1]$ to minimise the expected cumulative discounted cost from any initial state $s \in \mathcal{S}$, which is the *value-function* of state $s$ (alternatively one could work with rewards and maximisation)

$$V_\pi(s) = \mathbb{E}_\pi \left[ \sum_{k=0}^{\infty} \lambda^k c_{t+k} \mid S_t = s \right]. \tag{1}$$

In the above, $\mathbb{E}_\pi$ is the expectation under given policy $\pi$, the current time-step is $t$ and the cost at future step $t + k$ is $c_{t+k} = c(S_{t+k}, A_{t+k})$. The randomness is due to the user choice behaviour. We consider a *stationary policy* $\pi$, which is a distribution over actions given the current state. It does not depend on time $t$. This is a *randomised* policy in general,

$$\pi_s(\omega) := \mathbb{P}^\pi \left[ A_t = \omega \mid S_t = s \right], \qquad \omega \in \mathcal{A}(s). \tag{2}$$

If the mass is concentrated on a single slate-action $\omega$, the policy is called *deterministic* and we denote it by $\pi_s^d$ (or just $d$). Given a state $s$, it holds $\sum_{\omega \in \mathcal{A}(s)} \pi_s(\omega) = 1$. Observe that we introduced an action space $\mathcal{A}(s)$ per state $s$, because for our recommender application the currently viewed item $s$ should not be included in the recommendation slate.

The *state-action function* $Q_\pi(s, \omega)$ of pair $(s, \omega) \in \mathcal{S} \times \mathcal{A}$ is the expected cumulative discounted cost, starting from state $s$, taking action $\omega$ and following policy $\pi$,

$$Q_\pi(s, \omega) = \mathbb{E}_\pi \left[ \sum_{k=0}^{\infty} \lambda^k c_{t+k} \mid S_t = s, A_t = \omega \right]. \tag{3}$$

From Sutton and Barto [2018] and Puterman [1994] we know that the state-value functions satisfy the recursive system of *Bellman equations* (just policy $\pi$ evaluation here), $\forall (s, \omega) \in \mathcal{S} \times \mathcal{A}$

$$Q_\pi(s, \omega) = c(s, \omega) + \lambda \sum_{s' \in \mathcal{S}} \mathbb{P}\left[s' | s, \omega\right] \sum_{\omega' \in \mathcal{A}(s')} \pi_{s'}(\omega') Q_\pi(s', \omega') \tag{4}$$

$$= c(s, \omega) + \lambda \mathbb{E}_{s'} \left[ V_\pi(s') \mid S = s, A(s) = \omega \right], \tag{5}$$

where in the last equation we replaced with the value function in $s'$ because for stationary randomised policies it holds $V_\pi(s') = \sum_{\omega' \in \mathcal{A}(s')} \pi_{s'}(\omega') Q_\pi(s', \omega')$.

The $\mathbb{P}\left[s' | s, \omega\right]$ in (4) models the random user choice behaviour when visiting state $s$ and exposed to slate $\omega$, which is considered known in the MDP setting. Notice here, that the process is Markovian exactly because the user is Markovian, meaning that their choice is only based on the current state and action and not the past.

## 2.1 Item frequencies, transition probabilities, and state-item functions

For the decomposition we need to introduce some new marginal quantities to shift the analysis from slates to items. Since the policy $\pi$ is stationary and randomised, given state $s$ it randomly recommends among feasible slates, each containing a different set of items. Obviously, the item $j \in \mathcal{K}$ can appear in several action-slates.

**Definition 1.** *The frequency of a recommended item $j \in \mathcal{K}$ at state $s \in \mathcal{S}$, under policy $\pi$, is defined through the randomised probabilities of slate-actions in (2) as*

$$r_{s,j}^{\pi} := \mathbb{P}[A_t = \omega \in \mathcal{A}(s; \{j\})|S_t = s] = \sum_{\omega \in \mathcal{A}(s;\{j\})} \mathbb{P}^{\pi}[\omega|s] = \sum_{\omega \in \mathcal{A}(s)} \pi_s(\omega)\mathbf{1}(j \in \omega), \qquad (6)$$

where $\mathcal{A}(s; \{j\}) \subseteq \mathcal{A}(s)$ is the set of actions at state $s$, that necessarily include item $j$. The indicator function $\mathbf{1}(j \in \omega) = 1$ if $j$ is included in the slate, otherwise 0. What we call "frequency" is in fact the probability to randomly select some slate-action that includes item $j$, when at state $s$. It holds,

$$\sum_{j \in \mathcal{K}} r_{s,j}^{\pi} = \sum_{j \in \mathcal{K}} \sum_{\omega \in \mathcal{A}(s)} \pi_s(\omega)\mathbf{1}(j \in \omega) = N, \qquad \forall s \in \mathcal{S}, \qquad (7)$$

where we use the fact that each slate contains $N$ distinct items, i.e. there are no duplicates.

**Definition 2.** *The transition probability given item $j$ inside the recommended slate is defined as*

$$\mathbb{P}^{\pi}[s'|s, j] := \mathbb{P}[s'|s, \omega \in \mathcal{A}(s; \{j\})], \qquad \forall s \in \mathcal{S}, \ \forall j \in \mathcal{K}. \qquad (8)$$

For the transition probability given state $s$ and some action including item $j$ we prove three Properties:

**Property 1.** *The single-item transition probability depends on the policy $\pi$. It satisfies,*

$$\mathbb{P}^{\pi}[s'|s, j] = \sum_{\omega \in \mathcal{A}(s)} \mathbb{P}[s'|s, \omega]\mathbb{P}^{\pi}[\omega|s, j]\mathbf{1}(j \in \omega). \qquad (9)$$

*Proof.* We can write the slate as $\omega = (\omega_{-j}, j)$ which contains item $j$ and $\omega_{-j}$ are the remaining $N - 1$ entries. It holds due to conditioning, that

$$\mathbb{P}[s'|s, \omega] = \mathbb{P}[s'|s, (\omega_{-j}, j)] \quad = \quad \frac{\mathbb{P}^{\pi}[s', \omega_{-j}|s, j]}{\mathbb{P}^{\pi}[\omega_{-j}|s, j]} = \frac{\mathbb{P}^{\pi}[s', \omega|s, j]}{\mathbb{P}^{\pi}[\omega|s, j]},$$

where the superscript $\pi$ is included, because $\mathbb{P}^{\pi}[\omega|s, j]$ depends on the policy $\pi$. Using this expression,

$$\sum_{\omega \in \mathcal{A}(s;\{j\})} \mathbb{P}^{\pi}[s', \omega|s, j] \quad = \quad \sum_{\omega \in \mathcal{A}(s;\{j\})} \mathbb{P}[s'|s, \omega]\mathbb{P}^{\pi}[\omega|s, j].$$

Summing at the left-hand side over all $\omega$ that contain $j$, we get the marginal $\mathbb{P}^{\pi}[s'|s, j]$. $\qquad \square$

**Property 2.** *The single-item transition probability is a marginal probability of $\mathbb{P}[s'|s, \omega]$, and it holds*

$$\sum_{\omega \in \mathcal{A}(s)} \pi_s(\omega)\mathbf{1}(j \in \omega)\mathbb{P}[s'|s, \omega] = r_{s,j}^{\pi}\mathbb{P}^{\pi}[s'|s, j] \qquad \forall s \in \mathcal{S}, \ \forall j \in \mathcal{K}. \qquad (10)$$

*Proof.* It holds that $\pi_s(\omega) := \mathbb{P}^{\pi}[\omega|s]$. We can use the conditional probability formula

$$\sum_{\omega \in \mathcal{A}(s)} \pi_s(\omega)\mathbf{1}(j \in \omega)\mathbb{P}[s'|s, \omega] \quad = \quad \sum_{\omega \in \mathcal{A}(s)} \mathbb{P}[s', \omega|s]\mathbf{1}(j \in \omega)$$

$$= \mathbb{P}[s', \omega \in \mathcal{A}(s; \{j\})|s] \quad = \quad \mathbb{P}[s'|s, \omega \in \mathcal{A}(s; \{j\})] \cdot \mathbb{P}[\omega \in \mathcal{A}(s; \{j\})|s]$$

$$\stackrel{Def.2, Def.1}{=} \mathbb{P}^{\pi}[s'|s, j]r_{s,j}^{\pi}. \quad \square$$

**Property 3.** *If $r_{s,j}^{\pi} > 0$, then $\mathbb{P}[s'|s, j]$ is a probability mass function, $\sum_{s' \in \mathcal{S}} \mathbb{P}^{\pi}[s'|s, j] = 1$.*

*Proof.* Using Definition 2 and Definition 1 we can write

$$\sum_{s' \in \mathcal{S}} \mathbb{P}^{\pi}[s'|s, j] \quad \stackrel{Def.2}{=} \quad \sum_{s' \in \mathcal{S}} \mathbb{P}[s'|s, \omega \in \mathcal{A}(s; \{j\})] = \sum_{s' \in \mathcal{S}} \frac{\mathbb{P}[s', \omega \in \mathcal{A}(s; \{j\})|s]}{\mathbb{P}[\omega \in \mathcal{A}(s; \{j\})|s]}$$

$$= \quad \frac{1}{\mathbb{P}[\omega \in \mathcal{A}(s; \{j\})|s]} \sum_{s' \in \mathcal{S}} \sum_{\omega \in \mathcal{A}(s)} \mathbf{1}(j \in \omega)\mathbb{P}[s', \omega|s]$$

$$\stackrel{Def.1}{=} \quad \frac{1}{r_{s,j}^{\pi}} \sum_{\omega \in \mathcal{A}(s)} \pi_s(\omega)\mathbf{1}(j \in \omega) \sum_{s' \in \mathcal{S}} \mathbb{P}[s'|s, \omega] = \frac{1}{r_{s,j}^{\pi}}r_{s,j}^{\pi}1. \quad \square$$

**Definition 3.** *The* marginal *cost-item function $c^\pi(s, j)$ that depends on policy $\pi$ is defined as*

$$r_{s,j}^\pi c^\pi(s, j) := \sum_{\omega \in \mathcal{A}(s)} \pi_s(\omega) c(s, \omega) \mathbf{1}(j \in \omega), \qquad \forall s \in \mathcal{S}, \ \forall j \in \mathcal{K}. \tag{11}$$

*In the special case that the cost is just a function of the current state, $c^\pi(s, j) = c(s), \forall j \in \mathcal{K}$.*

Finally, we give the following special definition for the state-item function $Q_\pi(s, j)$:

**Definition 4.** *The state-item function $Q_\pi(s, j)$ is defined from the state-action functions $Q_\pi(s, \omega)$ as*

$$r_{s,j}^\pi Q_\pi(s, j) := \sum_{\omega \in \mathcal{A}(s)} \pi_s(\omega) Q_\pi(s, \omega) \mathbf{1}(j \in \omega) \qquad \forall s, j \in \mathcal{S}. \tag{12}$$

Notice that this definition is different from what would be the most natural one, i.e. to be the value-function starting from state $s$, and taking some initial slate-action that necessarily includes item $j$ and following policy $\pi$. The Definition 4 is in fact a marginal quantity, i.e., the *expectation over all state-value functions that include item $j$ normalised by the frequency $r_{s,j}^\pi > 0$.*

Notice here that for items $j$ such that $r_{s,j}^\pi = 0$, the state-item functions $Q_\pi(s, j)$ are not well defined. We will see next that this is not actually a problem for the decomposition.

## 2.2 Decomposed Bellman equations

**Theorem 1.** *[SlateFree-MDP] The Bellman equations in (4) for state-action functions with slates $Q_\pi(s, \omega)$, are equivalent to the following system of equations with state-item functions $Q_\pi(s, j)$ from Def. 4, cost-item functions from Def. 3, and transition probability given some item $j$ from Def. 2*

$$Q_\pi(s, j) = c^\pi(s, j) + \lambda \sum_{s' \in \mathcal{S}} \mathbb{P}^\pi[s'|s, j] \left( \frac{1}{N} \sum_{k \in \mathcal{K}} r_{s',k}^\pi Q_\pi(s', k) \right), \qquad \forall s \in \mathcal{S}, \ \forall j \in \mathcal{K}. \tag{13}$$

*If the cost is a function of just the current state, we replace in the above by $c^\pi(s, j) = c(s)$, so that the cost does not depend on the policy.*

*Proof.* We multiply both sides of (4) by $\pi_s(\omega)\mathbf{1}(j \in \omega)$ and sum over all feasible slate-actions $\omega$,

$$\sum_{\omega \in \mathcal{A}(s)} \pi_s(\omega) \mathbf{1}(j \in \omega) Q_\pi(s, \omega) = \sum_{\omega \in \mathcal{A}(s)} c(s, \omega) \pi_s(\omega) \mathbf{1}(j \in \omega) +$$
$$+ \ \lambda \sum_{\omega \in \mathcal{A}(s)} \pi_s(\omega) \mathbf{1}(j \in \omega) \sum_{s' \in \mathcal{S}} \mathbb{P}[s'|s, \omega] V_\pi(s').$$

Then, we replace the left-hand side by the function Definition 4, the first term of the right-hand side by the cost-item Definition 3 and in the second term we use Property 2, to find

$$r_{s,j}^\pi \left( Q_\pi(s, j) - c^\pi(s, j) - \lambda \sum_{s' \in \mathcal{S}} \mathbb{P}^\pi[s'|s, j] V_\pi(s') \right) = 0. \tag{14}$$

Now, use the equality $V_\pi(s') = \sum_{\omega \in \mathcal{A}(s')} \pi_{s'}(\omega) Q_\mu(s', \omega)$, multiply it from both sides by $\mathbf{1}(k \in \omega)$ and sum over $k \in \mathcal{K}$. We use the fact that the slate size is $N$ and again Definition 4, so we get

$$\sum_{k \in \mathcal{K}} V_\pi(s') \mathbf{1}(k \in \omega) = \sum_{k \in \mathcal{K}} \sum_{\omega \in \mathcal{A}(s')} \pi_{s'}(\omega) Q_\pi(s', \omega) \mathbf{1}(k \in \omega) \Rightarrow V_\pi(s') = \frac{1}{N} \sum_{k \in \mathcal{K}} r_{s',k}^\pi Q_\pi(s', k).$$

By replacing the above expression for $V_\pi(s')$ in (14) we get the expression in (13), as long as $r_{s,j}^\pi > 0$ for the $(s, j)$ pair. In the case that $r_{\tilde{s},\ell}^\pi = 0$ for some pair $(\tilde{s}, \ell)$, notice that regardless of its value $Q_\pi(\tilde{s}, \ell) < \infty$, it will always contribute $r_{\tilde{s},\ell}^\pi Q_\pi(\tilde{s}, \ell) = 0$ when found at the right-hand side of (13), hence the pairs with zero frequencies do not affect the equations of others. For their own state-item value, any solution $Q_\pi(\tilde{s}, \ell) - c(\tilde{s}) - \lambda \sum_{\tilde{s}' \in \mathcal{S}} \mathbb{P}^\pi[\tilde{s}'|\tilde{s}, \ell] V_\pi(\tilde{s}') = \kappa < \infty$ satisfies (14), hence also the one for $\kappa = 0$. This way we result in the validity of (13) for any possible state-item pair. $\qquad \square$

## 2.3 Optimality equations

We know from Puterman [1994, Prop.6.2.1] that the discounted MDPs always have a stationary deterministic optimal policy. We denote from now on deterministic policies by index $d$ (we may omit $\pi$) and the optimal policy by $d^*$ (we may omit $d$). Then by definition,

$$\pi_s^d(\omega) = \begin{cases} 1, & \text{for a unique slate } \omega^d(s) \in \mathcal{A}(s) \\ 0, & \forall \omega \neq \omega^d(s) \text{ and } \omega \in \mathcal{A}(s) \end{cases}. \tag{15}$$

A special case is when we follow the optimal deterministic policy, so that

$$\pi_s^*(\omega) = \begin{cases} 1, & \text{for } \omega^*(s) = \arg\min_{\omega \in \mathcal{A}(s)} Q_{d^*}(s,\omega) \\ 0, & \text{otherwise} \end{cases}. \tag{16}$$

When more than one action-slates have the minimum $Q(s,\omega)$ ties are broken arbitrarily. For the deterministic and optimal policies the value function starting from state $s$ is equal to

$$V_d(s) = \sum_{\omega \in \mathcal{A}(s)} \pi_s^d(\omega) Q_d(s,\omega) = Q_d(s,\omega^d(s)) \overset{opt.}{\Rightarrow} V_{d^*}(s) = \min_{\omega \in \mathcal{A}(s)} Q_{d^*}(s,\omega). \tag{17}$$

**Theorem 2.** *[Optimal SlateFree-MDP] The Bellman optimality equations for slates are equivalent to the following system of equations with state-item functions from Def. 4, cost-item functions from Def. 3, and transition probability given some item $j$ from Def. 2, under the optimal policy $\pi = d^*$*

$$Q_{d^*}(s,j) = c^{d^*}(s,j) + \lambda \sum_{s' \in \mathcal{S}} \mathbb{P}^{d^*}[s'|s,j] \min_{\ell \in \mathcal{K}} Q_{d^*}(s',\ell), \qquad \forall s \in \mathcal{S}, \forall j \in \mathcal{K} \tag{18}$$

*and it holds $Q_{d^*}(s,j) = V_{d^*}(s)$, $\forall j \in \omega^*(s)$ inside the optimal slate. Also, $c^{d^*}(s,j) = c_d(s,\omega^*(s))$ for $j$ in the optimal slate. For $\ell \notin \omega^*(s)$ the cost $c^*(s,\ell)$ can be any convex combination of the slate-costs $c(s,\omega)$, for $\omega \in \mathcal{A}(s;\{\ell\})$. For state-only dependent cost, we replace by $c^{d^*}(s,\omega^*(s)) = c(s)$.*

*Proof.* For deterministic (and optimal) policies the quantities in Section 2.1 related to items become:
1. *Item-frequencies* (from Definition 1)

$$r_{s,j}^d = \begin{cases} 1, & \forall j \in \omega^d(s) \\ 0, & \text{otherwise} \end{cases}. \tag{19}$$

2. *Transition probability* (from Definition 2):

$$\mathbb{P}^d[s'|s,j] = \mathbb{P}[s'|s,\omega^d(s)], \qquad \forall j \in \omega^d(s), \tag{20}$$

meaning that the transition probability given some item in the slate, is equal to the transition probability given the whole information about the slate. 3. *Cost* – if it depends on the action $\omega$ (from Definition 3):

$$c_d(s,j) = c_d(s,\omega^d(s)), \qquad \forall j \in \omega^d(s) \tag{21}$$

4. *State-item function* (from Definition 4):

$$Q_d(s,j) = Q_d(s,\omega^d(s)), \qquad \forall j \in \omega^d(s). \tag{22}$$

In words, given state $s$, all items included in the deterministic (resp. optimal) slate have the same state-item function value, equal to that of the whole slate.

**Lemma 1.** *For any stationary policy $d$ (and also the optimal $d^*$), it holds that $Q_d(s,j) = V_d(s)$, $\forall j \in \omega^d(s)$. Specifically for the optimal,*

$$Q_{d^*}(s,j) = \min_{\omega \in \mathcal{A}(s)} Q_{d^*}(s,\omega), \qquad \forall j \in \omega^*(s) \tag{23}$$

*Proof of Lemma 1.* For stationary deterministic policies we have from (22) that $Q_d(s,j) = Q_d(s,\omega^d(s))$, $\forall j \in \omega^d(s)$. It also holds from (17) that $V_d(s) = Q_d(s,\omega^d(s))$. $\square$

We now continue to the proof of Theorem 2. Applying the optimal deterministic policy to the state-action equations for item-frequencies from Theorem 1 (see formulation in 14) we get, ($c^* := c^{d^*}$)

$$Q_{d^*}(s,j) = c^*(s,j) + \lambda \sum_{s'} \mathbb{P}^{d^*}[s'|s,j] V_{d^*}(s') \overset{(17)}{=} c^*(s,j) + \lambda \sum_{s'} \mathbb{P}^{d^*}[s'|s,j] \min_{\omega \in \mathcal{A}(s)} Q_{d^*}(s',\omega).$$

From (23) in Lemma 1 it holds that $Q_{d^*}(s',j) = V_{d^*}(s')$, $\forall j \in \omega^*(s')$. Then, necessarily $Q_{d^*}(s',j) \leq Q_{d^*}(s',k)$, $\forall k \notin \omega^*(s')$, otherwise $V_{d^*}(s')$ would not be the optimal value. In other words, $V_{d^*}(s') = \min_{\ell \in \mathcal{K}} Q_{d^*}(s',\ell)$. From (11) we get (21) $c^*(s,j) = c^{d^*}(s,\omega^*(s))$ for all $j \in \omega^*(s)$. For $\ell \notin \omega^*(s)$, we know that $r_{s,\ell}^* \to 0$ and $\pi_s(\omega) \to 0$ for all $\omega \in \mathcal{A}(s;\{\ell\})$, so that from (11) $c^*(s,\ell)$ can be any convex combination of the slate-costs $c(s,\omega)$, for $\omega \in \mathcal{A}(s;\{\ell\})$. $\square$

## 3 Decomposed SARSA and Q-learning for slate actions

Consider a sequence of states, slate-actions and costs over discrete time-slots $t = 1, 2, \ldots$ as $(S_1 = s, A_1 = \omega, c_1 = c(s), S_2 = s', A_2 = \omega', c_2 = c(s'), \ldots)$. The transition from state $S_1$ to $S_2$ depends on the slate recommended by the agent, and the unknown user behaviour to select one of the items in the slate; the user is allowed to disregard the slate and select another item of their own preference. The vanilla SARSA Sutton and Barto [2018] is an on-policy $TD(0)$ method, which updates the state-action values $Q(s_t, \omega_t)$ as follows

$$Q(s_t, \omega_t) \quad = \quad Q(s_t, \omega_t) + \gamma \left[ c(s_t, \omega_t) + \lambda Q(s_{t+1}, \omega_{t+1}) - Q(s_t, \omega_t) \right]. \tag{24}$$

The slate-actions $\omega_{t+1}$ can follow the $\epsilon$-greedy exploration policy. We denote it by $\pi^\epsilon$; based on this, the greedy slate $\omega^*(s)$ that minimises $Q(s, \omega)$ is chosen with probability $1 - \epsilon$ and a uniformly random slate $\omega \in \mathcal{A}(s)$ is chosen with probability $\epsilon$. This implementation requires per state $s \in \mathcal{S}$, all $\binom{K}{N}$ combinations of Q-values stored in memory. Furthermore, all these combinations need to be traversed when searching for the minimum in the greedy step.

**SlateFree updates.** We can use the decomposition of the Bellman equations in Theorem 1, to propose a `SlateFree-SARSA` policy. We remind that a state-action pair $(s, \omega)$ corresponds to $N$ state-item pairs $(s, j)$ one per $j \in \omega$. The update can be written based on (13),

$$[\texttt{SlateFree} - \texttt{SARSA}] \qquad \text{For all } N \text{ items in the slate } j \in \omega_t :$$

$$Q(s_t, j) \quad = \quad Q(s_t, j) + \gamma \left[ c^\epsilon(s_t, j) + \lambda \frac{1}{N} \sum_{k \in \omega_{t+1}} Q(s_{t+1}, k) - Q(s_t, j) \right] \tag{25}$$

For the $\epsilon$-greedy policy, each time state $s_t$ is visited, the transition to $s_{t+1}$ is sampled from the unknown transition probability $\mathbb{P}^\epsilon[s_{t+1}|s_t, j]$, which depends on user preferences, but also on the policy $\epsilon$-greedy, which can change over time in the transient regime. The frequencies $r^\epsilon_{s', k}$ in (13) do not appear above, because the new action batch $\omega_{t+1}$ is a sample of the policy $\pi^\epsilon$ (and the frequencies $r^\epsilon$). In fact it can be easily shown that $\sum_{k \in \omega_{t+1}} Q(s_{t+1}, k)/N$ is just a one-sample *unbiased estimator* of $\sum_{k \in \mathcal{K}} r_{s', k} Q(s', k)/N$. The cost $c^\epsilon(s_t, j)$ also depends on the $\epsilon$-greedy; in the special case that it depends on the state only, we replace $c^\epsilon(s_t, j) = c(s_t)$, otherwise the cost per item will evolve over time and needs to be recalculated using Def. 3, keeping track of $\tilde{r}^\epsilon$, $\tilde{\pi}$ estimators.

Similar to SARSA, we can introduce a decomposed version of the Q-learning algorithm (Watkins and Dayan [1992]), which is an off-policy $TD(0)$ method, where the $Q$ functions are updated based on the optimal action policy, although the actions may follow some other (say $\epsilon$-greedy) policy. Then as above, we can use Theorem 2 to propose the updated step of the state-item functions following (18),

$$[\texttt{SlateFree} - \texttt{Q}] \qquad \text{For all } N \text{ items in the slate } j \in \omega_t :$$

$$Q(s_t, j) \quad = \quad Q(s_t, j) + \gamma \left[ c(s_t, \omega_t) + \lambda \min_{\ell \in \mathcal{K}} Q(s_{t+1}, \ell) - Q(s_t, j) \right]. \tag{26}$$

In the special case that it depends on the state only, we replace by $c(s_t, \omega_t) = c(s_t)$. The implementation of `SlateFree` (both -SARSA and -Q variations) requires per state $s$ at most $K$ values stored in memory (if we avoid self-loops, then recommending the same item is not an option).

**Finding the best slate.** In the exploitation phase of the $\epsilon$-greedy policy, we need to decide which $N$-slate is optimal. We are given, however, for each state $s$, not the state-action values $Q(s, \omega)$ but rather the $K$ state-item values $Q(s, j)$. Since we are looking for a stationary deterministic optimal policy, then we can apply the results from Section 2.3. Specifically, we have proved in Lemma 1 that $Q_{d^*}(s, j) = \min_{\omega \in \mathcal{A}(s)} Q_{d^*}(s, \omega), \forall s \in \omega^*(s)$, meaning that all state-item values will be equal, for the items included in the optimal slate (or more generally in the slate of the deterministic policy). Hence, we need only select in the greedy phase, the $N$ items with smallest $Q(s, j)$ values, both in the -SARSA and -Q version of `SlateFree`.

The update steps in (25) and (26) have important novelties compared to alternatives, as in e.g. Ie et al. [2019, eq.14, 15]. They are strictly model-free and do not need any prior knowledge over the environment. Also, costs that depend on both state and action are allowed. Hence, `SlateFree` uses $N$ parallel updates to learn any stationary environment over time, using a more compact Q-function representation, compared to the non-decomposed vanilla-SARSA and Q-learning methods. Convergence to the optimal is empirically verified in practice, but yet not provably guaranteed, due to the dependence of the costs and transition probabilities per item in the learned policy.

# 4 Numerical evaluation

In this section, we evaluate numerically the performance of `SlateFree` (both -*Q* and -*SARSA* variations), against two methods from the literature: (i) the standard Q-learning and SARSA tabular method (called *Vanilla-Q* and *Vanilla-SARSA*, where all possible $\binom{K}{N}$ action-slate combinations are accounted for, each having its own Q-value per state; furthermore against (ii) the proposed in Ie et al. [2019] method *SlateQ* with greedy slate selection in exploitation phase. Our environment is a recommendation system where a user starts their viewing episode from a certain item, and the system recommends a slate of $N$ (unordered) items, excluding the currently viewed item. Each episode has average length $(1 - \lambda)^{-1}$ steps. The experience is repeated over $T_{epis} \geq 1$ episodes. The great challenge is to test the `SlateFree` method on user behaviours whose solution needs to be combinatorially searched. We evaluate three such artificial user choice models:

User-1 The user has a fixed retention probability $\alpha \in [0, 1]$ per step to select one of the $N$ recommended items uniformly at random, and $(1 - \alpha)$ probability to disregard the recommendations and select on their own for one of the $K$ library items uniformly at random.

User-2 The user has a set $\mathcal{X}$ of undesired items that they would never click on. Then, this behaviour is similar to user-1, with the difference that user-2 will choose either from the recommended items (with probability $\alpha$), or from the whole library (with $1 - \alpha$) one item uniformly at random among all items, ignoring in both cases those items in set $\mathcal{X}$.

User-3 The user has a set $\mathcal{Y}$ of must-include global items. This user does not follow a retention probability $\alpha$. Instead, they will select among the recommended items at random, as long as at least one item from $\mathcal{Y}$ is included in the recommendation slate.

*A. Small scenario.* We first study a small size scenario with $K = 10$ and $N = 4$. The number of possible combinations per state is $\binom{K-1}{N} = 126$, where we exclude recommendation of the currently viewed item. With `SlateFree` we get a reduction in memory for the Q-table from $10 \times 126$ to $10 \times 9$. The costs per item are all high $20 + z_i$, where $z_i \sim Uniform(0, 4)$, but there are four items with lower cost, namely $c_0 = 5 + z_0$, $c_1 = 0 + z_1$, $c_7 = 4 + z_7$ and $c_9 = 8 + z_9$ (remember it is a minimisation problem). The discount is fixed in all experiments to $\lambda = 0.85$. For user-1 and -2, the retention is $\alpha = 0.75$. For user-2 the exclusion set $\mathcal{X} = \{0, 1, 8\}$. For user-3 we select $\mathcal{X} = \mathcal{Y}$ as in user-2. The learning rate is $\gamma = 0.004$ and the $\epsilon$-greedy GLIE strategy has fixed $\epsilon = 0.05$ for the exploration probability. We consider as number of episodes for the evaluation $T_{epis} = 600K$. Each episode is a walk of the user on the library of $K$ items, with mean length $(1 - \lambda)^{-1} = 6,67$ views. The evaluation for the three types of users and $T_{epis} = 600K$ is shown in Fig. 1 (TOP-row). We use on the x-axis logarithmic scale of episodes. The value per episode has high variance and so we smooth the results within a window of 200 episodes. `SlateFree-Q` and `-SARSA` converge for any user type in around $10K$ episodes, an order of magnitude faster than the Vanilla-Q and Vanilla-SARSA. Also, the average value after convergence is the same for the two methods, indicating that `SlateFree` converges to the optimal value function. Both `SlateFree` and SlateQ converge to the optimal value, but we will see this is not true for larger catalog and dimension instances. Our method shows faster and steeper convergence than SlateQ in all users, because SlateQ updates the item-Q values only for the single selected item, whereas `SlateFree` for all $N$ items included in the slate.

*B. Larger Scenario.* Next we evaluate the convergence in a more difficult scenario with $K = 100$ and $N = 10$, which corresponds to $\binom{99}{10} \approx 15 \cdot 10^{12}$ combinations. This problem is not tractable for Vanilla-Q or Vanilla-SARSA. Hence, we only show results for `SlateFree-Q`, `SlateFree-SARSA` and SlateQ in Fig. 1 (BOTTOM-row). The value per episode has high variance and so we smooth the results within a window of 1000 episodes. Now, both `SlateFree-Q` and `-SARSA` can solve all three user cases within $\approx 500K$ episodes, whereas SlateQ seems to learn and improve over time, but cannot solve for any user, at least within the $T_{epis} = 1M$ episodes.

*C. Insensitivity in $N$.* Our evaluations show that the convergence time given some library size $K$ becomes almost insensitive to the dimension size $N$. To illustrate this, we simulate user-3 for a catalog size $K = 10$ and various sizes of $N \in \{1, 2, 3, 4, 5\}$. The results are illustrated in Fig. 2 (left). One can observe that surprisingly the slowest converging curve is for $N = 1$, whereas for the higher $N$ almost all curves converge before $T_{epis} = 10K$. The reason for the poor convergence behaviour of $N = 1$ is probably due to the fact that each step in the episode contributes a single update of the state-item functions, whereas for $N > 1$ the multiple parallel updates accelerate the

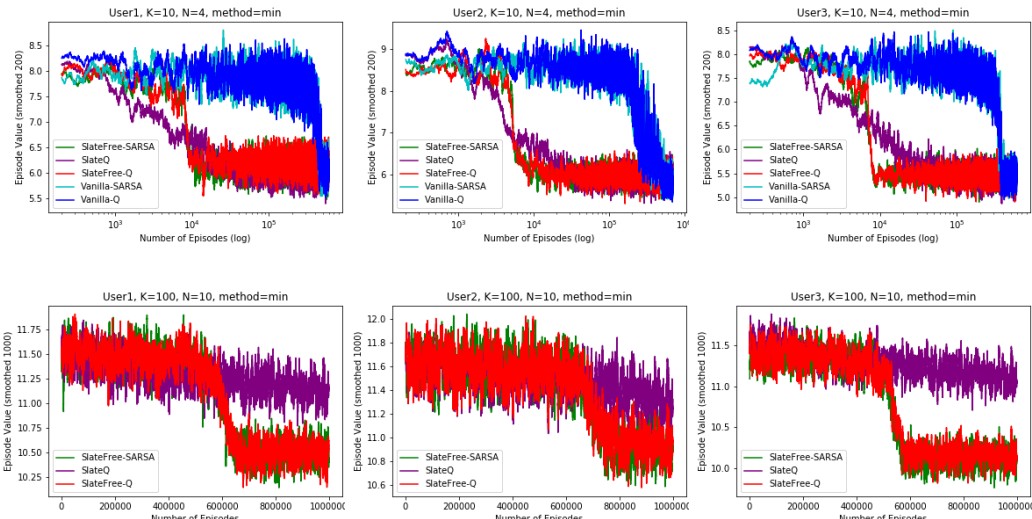

Figure 1: (TOP-row) Value function user-1 (left), user-2 (centre), user-3 (right); library $K = 10$, dimension $N = 4$, episodes $600K$, methods: `SlateFree-Q`, `SlateFree-SARSA`, `SlateQ`, `Vanilla-*`. (BOTTOM-row) Value function user-1 (left), user-2 (centre), user-3 (right); library $K = 100$, dimension $N = 10$, episodes $1M$, methods: `SlateFree-Q`, `SlateFree-SARSA`, and `SlateQ`.

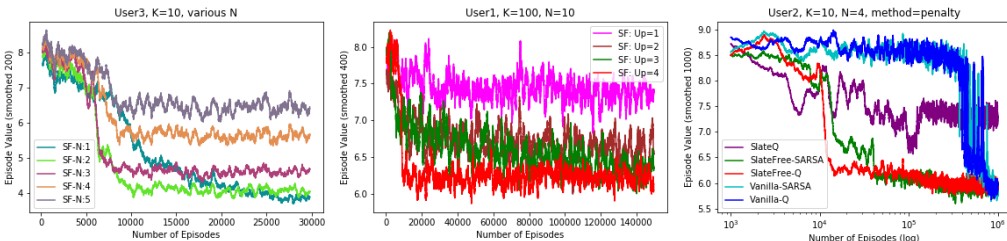

Figure 2: (left) Insensitivity in $N$, (centre) Number of Q-updates, (right) Slate-dependent cost.

process considerably. This is the same reason why `SlateFree-Q` in Fig. 1 shows a steeper learning curve compared to SlateQ, where the latter updates a single state-item function per step.

*D. Effect of parallel updates.* We investigate the role of $N$ parallel updates in the convergence of the `SlateFree-Q` algorithm. More specifically, for user-1, and the small scenario $K = 10$ and $N = 4$ we evaluate the algorithm using a different number of updates per step at each evaluation. Aside the proposed algorithm which updates all four items in the slate per step, in the others we allow three items per step, two items, and finally a single item to update. We plot our results in Fig. 2 (centre). We observe that the complete method with all four updates converges in $10K$ episodes already (shown in red). For three updates per step, the method seems to converge (green curve) to a value close to the optimum, albeit very slowly. For two and a single update (brown and pink curves) we observe that the method gradually improves over the episodes but even in $150K$ events it has not converged to the optimum. To conclude, the plot shows that it is necessary to do all $N$ parallel updates per step for the method to converge to the best possible value, and fast.

*E. Dependence of cost on both state and action-slate.* We study now how the performance of `SlateFree` is affected when the cost depends on both the current state and the entire action-slate $c(s_t, \omega_t)$. Such an option is not supported by SlateQ Ie et al. [2019]. We now modify the cost so that a penalty $= 42$ is applied to all $Q(s_t, j)$ where $j \in \omega_t$ are the items participating in the recommended slate, whenever the user does not follow (rejects) the recommendation slate. Obviously this penalty is slate-dependent. We illustrate the performance of all methods in Fig. 2 (right). We observe that the decomposed `SlateFree` converges to the optimal solution for both `-Q` and `-SARSA` variations, same as Vanilla-Q and Vanilla-SARSA. SlateQ from Ie et al. [2019] fails to converge to the minimal value.

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

# A  Appendix

No appendix accompanies the submission. The 9 pages and extra references are self-sufficient. Two URLs (for Dropbox SlateFree Authors [2022b] and for Google Colab SlateFree Authors [2022a]) link to the code used for the numerical evaluations.

