# OpenReview forum: "SlateFree: a Model-Free Decomposition for Reinforcement Learning with Slate Actions"
_NeurIPS.cc/2022/Conference — NeurIPS 2022 Submitted_

### Official Review · Reviewer_MKZi · 2022-07-10

**Rating:** 7
**Confidence:** 4
**Soundness:** 4 excellent
**Presentation:** 4 excellent
**Contribution:** 3 good

**Summary:**

This paper considers the problem of N items recommendation system which is a intractable due to the combinatorial nature of the problem. The authors introduce a slate-MDP for solving such problems. Through theoretical analysis, the authors show that the slate-MDP can be decomposed using just item-related Q function per state. This paper also proposes a SlateFree algorithm which is shown to be  insensitive to the slate size.
Some numerical experiment results are presented to show the effectiveness of the proposed algorithm. The proposed algorithm converges  faster compared to the SOTA algorithms.


**Questions:**

The second assumption seems to be essential. By assuming the reward is only a function of individual items, the combinatorial problem is simplified. I can understand this assumption is essential for getting the theorems. The numerical evaluations shows this assumption can be raised. But a similar result does not hold for SlateQ algorithm. Can you provide more details about this comparison? Both the theorems of these algorithm assume the independency.
In this experiment, the SlateQ algorithm converges faster than the proposed one for User1 setup. What is the reason that SlateQ performs better when user select the item by sampling from a bernoulli distribution? It will be great if you can provide more analysis for the experiment results.


**Limitations:**

Yes. The authors provide adequate information about this.

**Strengths And Weaknesses:**

The problem considered in this paper is promising since it is a generalization of the standard recommendation system problem. Due to the size of the action space is extremely large, it is important to have a time efficient algorithm. The proposed algorithm is shown to achieve good performance for such problems. Sufficient analysis is provided to support the theorems in this paper. The idea of decomposing the slate-based Q function into individual item-based Q function makes sense to me. Overall, this paper is well written and easy to follow.

---

> ### Author Response · Authors · 2022-07-31
> **Response to reviewer MKZi**
>
> We thank the reviewer for the critical reading of our manuscript.  Since more than one reviewer raised questions about the generality of the decomposition related to the state-dependent cost assumption, we first provide a general answer and then respond to the reviewer's comments in detail. Our method actually solves the general problem without independence assumptions and below are the arguments.
>
> Two important remarks - TO ALL REVIEWERS:
>
> [a] In the rebuttal version we have completely removed the assumption about state dependent cost. This was included in the submitted version as a strong -- but unnecessary -- assumption in our analysis. Besides, a similar assumption was used in [Ie et al, IJCAI'19]. As it turns out, this was a very conservative choice for SlateFree: we have found out that the proofs of Theorem 1 and Theorem 2 do not actually need such an assumption (see rebuttal version, modifications in blue). Instead, we need to introduce in the Theorems the marginal cost-item from Definition 3 $c(s,j)$, and use equality (21) for deterministic policies. (Both the cost-item definition and eq.(21) were already presented in the first submitted version.) In this way we can easily generalize the proofs. The incorporation of the marginal cost-item $c(s,j)$ in the Theorems 1 and 2 slightly modifies the SlateFree-SARSA and -Q methods. Eventually, this explains why, when we use slate-dependent costs in the numerical evaluation section, the SlateFree version converges to the vanilla tabular-Q solution, see last subsection and Fig.2 (right).
>
> [b] Definition 2 is a probability mass function (see Property 3) that results as the marginal of the transition probability given the full slate. This is clearly shown in Property 1, and Property 2, where all the remaining recommendation entries apart from the one occupied by item j are taken in expectation. We do not make any independence assumptions here, we just do a transformation of the original distribution based on marginals.
>
> All these small modifications and comments can be found in the rebuttal version of our work marked with color blue.
>
> ------------------------------------------
> Specific to reviewer MKZi
> ------------------------------------------
> As explained above, the assumption made about the state-dependent cost in the original submission has been now removed in the rebuttal version. Hence the MDP-decomposition is general for any cost depending on both states and actions. This is why the SlateFree can solve the general problem in the last subsection of the numerical evaluations treating the whole content of the slate jointly.
>
> - As the reviewer well observed the cost assumption was not necessary and the SlateFree method works in the numerical evaluations also for costs that depend on the whole action-slate. We verify this in the rebuttal version, where both Theorems 1, 2 now are proved for general costs, and the SlateFree-SARSA and -Q have been modified to treat such costs.
>
> - Considering the comparison with SlateQ: SlateQ essentially uses item-Q values (as in our SlateFree method), but with a different definition. In their approach only one item is updated per step, the one that the user chooses out of the recommended slate. The SlateQ completely disregards the rest of the unselected items inside the recommendation slate. This is why for most cases, also for the Bernoulli distribution, our SlateFree converges faster than the SlateQ, because it updates all items in the recommended slate not just the one selected. We have included in blue in the rebuttal version a comment about this at the end of Section "4.A Small scenario".

---

### Official Review · Reviewer_8bDr · 2022-07-10

**Rating:** 4
**Confidence:** 5
**Soundness:** 3 good
**Presentation:** 2 fair
**Contribution:** 2 fair

**Summary:**

This paper studies the sequential recommendation problem and shows that the Q functions can be decomposed based on the state-dependent cost function assumption. Evaluation results are also presented in the paper to verify the efficiency of the algorithm.

**Questions:**

1. Line 124, reward→cost.
2. If $S=K,$ why not just use one notation?
3. I have a question about the model. From the definition from Sec.2, the state s is the currently viewed item. Can the viewed items change over time or do they remain the same available $K$ items? If the states can change, I didn’t see the formal definition of the state space. If not, what do you mean (lines 109–110) by saying "the user moves to state $s’$ by either picking one of the recommended items or selecting something else from the catalog?” Will the MDPs stop if the user picks a recommended item? If so, the MDP should include a terminal state, and all the proofs need to be reconsidered.
4. What is the case when K=10 and N=4, and you said the memory for the Q-table was reduced from 10x126 to 10x9 calculated?

**Limitations:**

I don't think this paper has any potential negative societal impact.

**Strengths And Weaknesses:**

**Strengths**
1. The sequential recommendation problem is interesting.

**Weaknesses**

1. The presentation is a little hard to follow.
2. The decomposition is only possible when the cost is state-dependent; no further discussions are included for other cases. For example, what if the Q function can be decomposed approximately, or is local convergence achievable in other sensorial?
3. The technical contributions seem limited.

---

> ### Author Response · Authors · 2022-07-31
> **Response to reviewer 8bDr**
>
> We thank the reviewer for the critical reading of our manuscript. Since more than one reviewer raised questions about the generality of the decomposition related to the state-dependent cost assumption, we first provide a general answer and then respond to the reviewer's comments in detail. Our method actually solves the general problem without independence assumptions and below are the arguments.
>
> Two important remarks - TO ALL REVIEWERS:
>
> [a] In the rebuttal version we have completely removed the assumption about state dependent cost. This was included in the submitted version as a strong -- but unnecessary -- assumption in our analysis. Besides, a similar assumption was used in [Ie et al, IJCAI'19]. As it turns out, this was a very conservative choice for SlateFree: we have found out that the proofs of Theorem 1 and Theorem 2 do not actually need such an assumption (see rebuttal version, modifications in blue). Instead, we need to introduce in the Theorems the marginal cost-item from Definition 3 $c(s,j)$, and use equality (21) for deterministic policies. (Both the cost-item definition and eq.(21) were already presented in the first submitted version.) In this way we can easily generalize the proofs. The incorporation of the marginal cost-item $c(s,j)$ in the Theorems 1 and 2 slightly modifies the SlateFree-SARSA and -Q methods. Eventually, this explains why, when we use slate-dependent costs in the numerical evaluation section, the SlateFree version converges to the vanilla tabular-Q solution, see last subsection and Fig.2 (right).
>
> [b] Definition 2 is a probability mass function (see Property 3) that results as the marginal of the transition probability given the full slate. This is clearly shown in Property 1, and Property 2, where all the remaining recommendation entries apart from the one occupied by item j are taken in expectation. We do not make any independence assumptions here, we just do a transformation of the original distribution based on marginals.
>
> All these small modifications and comments can be found in the rebuttal version of our work marked with color blue.
>
> ---------------------------------------------
> Specific to reviewer 8bDr
> ---------------------------------------------
>
> - We understand that the decomposition is not straightforward and have tried in the rebuttal version to include certain small additions (space permitting) to help the reader, see text in blue.
>
> To provide a bit of intuition about our method, since the SlateFree-Q algorithm will update the Q-values of all items which take part inside the slate, this will have an additive effect in the item Q-value each time an item appears in some slate. As a means of example: If the specific 4-slate (i,j,k,l) is recommended at state s, then updating item i value in Q(s,i) will provide information also about other slates where i appears (i,j,l,m) or (i,l,m,f) and which have not been tested before. Hence, item-Q values are updated more efficiently than in the vanilla-Q case.
>
> Regarding the reviewers questions:
>
> - (1.) He have corrected Line124.
>
> - (2.) The reviewer's comment about state-space S and catalog K being identical is valid. We use two different symbols because the state can be more general, e.g. a history of viewed items within a window of some size (e.g. the (say) five last viewed videos in Youtube). Then the state-space would be different than the catalog. We want also to consider such cases, and a comment is included in the rebuttal version (in blue).
>
> - (3.) In the MDP formulation we consider that the catalog is fixed, so that the viewed item will be one from the catalog K. The state-space is the catalog, but as we briefly mention in the rebuttal version, it can be all the combinations of m-consecutive viewed items.
>
> What we mean in lines 109-110 is that the user has the option either to select from the recommended slate one item, or completely ignore the slate and pick at random (going to the search bar and following their own preferences) one item from the general catalog. This is also shown in the User-examples of the numerical evaluation. The MDP does not stop when the user picks an item and we do not need to include a terminal state. Actually, the length of each session follows a Geometric distribution of parameter lambda (the discount). We have included in blue a modification in the text of the rebuttal version to clarify.
>
> - (4.) In the case K=10 and N=4 the tabular Q-learning needs to store per state all combinations of items from the catalog, excluding currently viewed item. So the number of these combinations is 9-over-4, i.e. binomial(9,4) = 126 per state, so the Q-table size is 10x126, where 10 is the number of catalog items (states). The SlateFree algorithm needs only 10x9 entries in the table of item Q-values, due to the decomposition we have performed. So, we obtain an important reduction in memory space.

---

> > ### Comment · Reviewer_8bDr · 2022-08-07
> > **Response to Authors**
> >
> > Thanks to the authors for their response and clarification.
> >
> > Removing the assumption of dependence on the state is nice. However, I guess for implementing the SlateFree algorithm, in particular for the case when the cost depends on both states and actions, it still needs to calculate $c(s,j)$ at each step. This calculation also needs a large space based on the definition of 3. I am not even sure how to do the calculation in a completely model-free and data-driven approach like Vinalia Q-learning or SARSA, because it appears that just the cost for the current $c(s, w)$ is insufficient.
> >
> > I also have a follow-up question with the definition of the slate-MDPs. The transition kernel is a function from $S\times A \rightarrow S'$. The action space is the unordered slate of the recommended items. How is the users' behavior modeled in the definition? Does that mean the users' response has nothing to do with the MDP? Then how does the recommendation system get feedback from users?
> >
> > I strongly recommend using different notations to differentiate between $Q(s, w)$ and $Q(s, j),$ as well as the definition of the cost function. It is very confusing.
> >
> > The reason why I asked how the 126 is calculated is because in the introduction you said K choose N but you did the K-1 choose N in the simulation section instead. It is preferable to maintain consistency. I also agree with reviewer CrKM that it would be quite helpful if you could include a simple example (maybe the Youtube example you mentioned) in the definition of the slate-MDPs.

---

> > > ### Author Response · Authors · 2022-08-08
> > > **Follow-up response to reviewer's 8bDr comments**
> > >
> > > We are grateful to the reviewer for the very interesting comments.
> > >
> > > - We understand the reviewer's worries in the case when the cost depends on both state and action. But, in fact, the situation is very simple:
> > >
> > > (case SARSA) Indeed, in the case of SlateFree-SARSA one needs to calculate $c(s,j)$ at each step. We have mentioned in Section 3 (page 7, paragraph under eq.(25) in blue) that the item-cost from Definition 3 needs to keep in memory estimates of the frequencies $r_{s,j}$ and $\pi_s(\omega)$ which are just the frequencies to recommend item $j$ when visiting state $s$ and the frequencies to recommend slate $\omega$ when visiting state $s$, respectively.  Obviously, these frequencies are not related to the user preferences, they are just related to the policy updates.
> > >
> > > (case Q) Luckily, as shown in eq.(26), for the SlateFree Q-updates we do not need to use the item-cost $c(s,j)$ in the updates, but rather the original slate-cost $c(s,\omega)$ at each step. This comes directly from the updated Theorem 2 in the rebuttal version. Hence for the SlateFree Q-updates one does not need to calculate anything new related to item-costs, just use the known slate-costs in the original formulation.
> > >
> > > - Considering the follow-up question about the definition of the slate-MDPs: The user's response determines the probability to transit to state $S'$ given current state and recommendation slate $S\times A$. This means, if the user watches video $S$ and is suggested slate $A$ of say three items (I_a, I_b, I_c), the probability distribution to go to next state S' depends on the user's behaviour. For example, a user might choose item I_a with probability 1. Some other user might chose between I_b, or I_c with probability 50-50. Another user might decide that the suggested items (I_a, I_b, I_c) are not interesting and can choose item I_d and I_e from the catalog (outside the recommendations) with probability 75-25. In general, the reaction of some user to the recommended slate is summarised in the transition probability to S', given the current state and the slate. This probability is sampled in the Q-updates eq.(25) and eq.(26) of our SlateFree method. The recommender gets feedback from users by the transitions from S to S'.  In the SlateFree, which updates all items in the slate per slot, we do not care exactly which item is the one the user will chose, we rather care which slate of items is the best to be recommended, so that the user will minimize the cost with their special behaviour.
> > >
> > > - Following the reviewer's comments we plan to update notation in the final version of our paper. We can use e.g. \tilde{Q} and \tilde{c} for the newly introduced item-values and item-costs.
> > >
> > > - Also, we agree with the reviewer that the text should be more consistent related to K or K-1 over N combinations and we will do the appropriate corrections in the final version.
> > >
> > > - We will definitely include the example requested by both reviewer 8dBR and CrKM in the final version of our work. We have confirmed this also in our response to the reviewer CrKM.
> > >
> > > We would like to thank again the reviewer 8dBR for the careful reading and the constructive comments.

---

### Official Review · Reviewer_CrKM · 2022-07-10

**Rating:** 4
**Confidence:** 3
**Soundness:** 2 fair
**Presentation:** 2 fair
**Contribution:** 2 fair

**Summary:**

This work proposes (under some assumptions) a novel approach to RL for slate based recommendations. In general, standard RL approaches are not tractable since the action space when considering slates of recommendations is combinatorially large.

Here, they introduce assumptions that allow them to decompose the problem into learning action-values on each item in a slate.

They test the approach on a simulated environment they designed.

**Questions:**

Isn't the assumption that the cost depends only on the state and not the action very unrealistic? $c(s, \omega)=c(s)$. In particular, in almost all recommendation problems the reward is likely to be something like the expected user engagement which is action dependent. I'm aware you relax this assumption in a specific numerical experiment (line 318), but this is still very different than the general case of action-dependence.

Related to the above point. Much of the interest in slate based approaches is that the value of recommending an item depends on other items in the slate. For example, if the state allows us to infer that the user is interested in either basketball or baseball, showing a (slate size 2) basketball recommendation and a baseball recommendation is probably a good slate. But showing two (potentially relevant) basketball videos is likely to suboptimal since it is less diverse so the additional value of the second basketball video is much lower. It seems like the approach proposed here does not address this problem of diversity and that the value of an item in a slate may depend on the other items in the slate?

Why use $\omega$ to denote the action instead of $a$?


**Limitations:**

I don't think this work discusses the limitations of this approach clearly. It would helpful to test on some other environments and explain more clearly the limits of the assumptions needs for this decomposition.


**Strengths And Weaknesses:**

Strengths:

RL for slate based recommendations is an interesting topic.

Weakness:

I found the decomposition proposed here difficult to follow. It would be helpful to try and provide more intuition behind this approach.

I have some concerns about the realism of the assumptions and whether its solving the main issues of interest in slate based recommendations (see questions).

The approach is only tested on a simple simulation tailored to this approach. There are a number of public (real) datasets that can be used for evaluating this approach using off-policy estimators or SlateQ released a simulation with their method. Testing on a broader range of environment (particularly with real data) would help better understand the useful and realism of this approach.

---

> ### Author Response · Authors · 2022-07-31
> **Response to reviewer CrKM**
>
> We thank the reviewer for the critical reading of our manuscript. Since more than one reviewer raised questions about the generality of the decomposition related to the state-dependent cost assumption, we first provide a general answer and then respond to the reviewer's comments in detail. Our method actually solves the general problem without independence assumptions and below are the arguments.
>
> Two important remarks - TO ALL REVIEWERS:
>
> [a] In the rebuttal version we have completely removed the assumption about state dependent cost. This was included in the submitted version as a strong -- but unnecessary -- assumption in our analysis. Besides, a similar assumption was used in [Ie et al, IJCAI'19]. As it turns out, this was a very conservative choice for SlateFree: we have found out that the proofs of Theorem 1 and Theorem 2 do not actually need such an assumption (see rebuttal version, modifications in blue). Instead, we need to introduce in the Theorems the marginal cost-item from Definition 3 $c(s,j)$, and use equality (21) for deterministic policies. (Both the cost-item definition and eq.(21) were already presented in the first submitted version.) In this way we can easily generalize the proofs. The incorporation of the marginal cost-item $c(s,j)$ in the Theorems 1 and 2 slightly modifies the SlateFree-SARSA and -Q methods. Eventually, this explains why, when we use slate-dependent costs in the numerical evaluation section, the SlateFree version converges to the vanilla tabular-Q solution, see last subsection and Fig.2 (right).
>
> [b] Definition 2 is a probability mass function (see Property 3) that results as the marginal of the transition probability given the full slate. This is clearly shown in Property 1, and Property 2, where all the remaining recommendation entries apart from the one occupied by item j are taken in expectation. We do not make any independence assumptions here, we just do a transformation of the original distribution based on marginals.
>
> All these small modifications and comments can be found in the rebuttal version of our work marked with colour blue.
>
> ---------------------------------------------
> Specific to reviewer CrKM
> ---------------------------------------------
>
> - The proposed decomposition is indeed not straightforward, but it is original; it makes use of marginal quantities (in cost, transition, Q-value) that we define and work with. These quantities do not constitute independence assumptions, they rather use the original model elements to reformulate the MDP, taking expectations. We have included in the rebuttal version certain short (space permitting) additions that may help more in clarifying this.
>
> To provide a bit of intuition about our method, since the SlateFree-Q algorithm will update the Q-values of all items which take part inside the slate, this will have an additive effect in the item Q-value each time an item appears in some slate. As a means of example: If the specific 4-slate (i,j,k,l) is recommended at state s, then updating item i value in Q(s,i) will provide information also about other slates where i appears (i,j,l,m) or (i,l,m,f) and which have not been tested before. Hence, item-Q values are updated more efficiently than in the vanilla-Q case.
>
> - The simulation setup is indeed simple with small catalog sizes, it does however serve sufficiently the purpose we want, that is to validate our decomposition, especially for small scenarios where the tabular-Q version is tractable and we can find the optimum. Note that in large real datasets no tabular-Q method is possible to work with, because the action-space would be immense making the problem intractable. Deep-RL should be used for large state-space rather than large action-space, hence other approximative methods such as amortized inference (Wiele et al 2020) should be applied. The decomposition we propose here is exact and can be used as a prior transformation in Slate problems, before applying more elaborate deep architectures like the ones mentioned. This is an interesting topic for future work.
>
> - We agree with the reviewer that the entire slate should be accounted for, and this is exactly what the SlateFree decomposition does using marginal quantities for transitions, Q-values and costs, rather than independence assumptions: It takes into account the content of all items (as the sports example given by the reviewer suggests). The item-transition probability, i.e. the transition from state s to s' given item j in the slate (see Def.2) is actually the marginal of the slate-transition probability; hence item-probabilities summarize information over all slates including item j, but they do not assume any independence.

---

> > ### Comment · Reviewer_CrKM · 2022-08-07
> > **Still concerned about practicality and generality**
> >
> > Firstly, thanks to the authors for responding substantively and updating their proofs to remove the most unrealistic assumption. The decomposition they propose is interesting.
> >
> > Because of this I am raising my score. I still do think the paper is a little hard to follow, and in particular it would be great to provide some simple examples (but I also acknowledge space is tight, maybe by moving some proofs to an appendix).
> >
> > My main remaining concern is the practicality of this approach. Many (most?) recommender systems have extremely large number of items. The simulation here is only to K=100, (many real recommender systems might be K=1e7 or more items) and this approach seems to require a lot of observations under a stationary policy (to avoid imposing a user behavior model like in SlateQ).
> >
> > I think it would be improve by evaluating on some external benchmark tasks, such as the simulation in SlateQ (and ideally a real recommender system although that may not be practical).

---

> > > ### Author Response · Authors · 2022-08-08
> > > **Response to reviewer CrKM concerns**
> > >
> > > We sincerely thank the reviewer for the constructive comments. Indeed it is a good idea to include a simple example that can provide intuition and clarity. In the final version of our paper we plan to move some proofs in the Appendix, as suggested, and include an example case of a sequential recommendation system and a user with preferences towards some category, e.g. sports. Using such scenario we will explain intuitively how the algorithm updates the Q-values and finds the optimal policy.
> > >
> > > Concerning the comment about the practicality of our approach: Current sequential recommendation systems already need to struggle with such large catalogs (K=1e7) as the reviewer mentions, and do so through a combination of deep reinforcement learning and policy gradients. Such variations are in essence value approximations of temporal difference updates coming from the basic Bellman equations.
> > >
> > > But, it is exactly these Bellman equations that we show in our work that can be written in a more compact way using our suggested SlateFree decomposition. This way we already show that our method can solve much larger problems than the ones treated by the original tabular Q-learning. Our method is the first step which establishes the decomposition and the new Q-learning algorithm.
> > >
> > > As a follow-up to this work we plan to extend the suggested SlateFree Q-learning algorithm in order to incorporate value approximation algorithms (using Deep RL and/or policy gradients) in order to support real world size catalogs (K=1e7) and also compare our method with benchmark tasks, like the simulation in SlateQ for very large datasets.
> > >
> > > Thank you very much again for your careful evaluation.

---

### Official Review · Reviewer_1QMW · 2022-07-11

**Rating:** 3
**Confidence:** 4
**Soundness:** 2 fair
**Presentation:** 3 good
**Contribution:** 2 fair

**Summary:**

This paper tackles the problem of sequential slate recommendations, where at each step the agent has to choose from a combinatorially large action set. It defines some assumptions for the structure of the MDP (called SlateFree-MDP) and then derives the Bellman equations for this setup. The modified Q-learning and SARSA algorithms then operate on these modified Bellman equations.

**Questions:**

- The definition of the state is not quite clear to me. If the state is the last-viewed item, the transition is the next viewed state under a given policy?
- What is the difference between the terms $P^\pi[\omega|s,j]$ and $P[\omega \in A(s;j)|s]$? Given that the policy depends only on the current state.
- How applicable are these structural assumptions to some real-world data/application?


**Limitations:**

The limitations are partially unaddressed as mentioned above, and the societal impact of using this on recommender systems is not discussed.


**Strengths And Weaknesses:**

Strengths:
- The decomposition of the original MDP results in significantly reduced computation (due to the independences introduced)
- Short proofs help understand the properties and definitions as the reader proceeds.
Weaknesses:
- The assumptions, particularly Definition 2 along with the state (only) dependent cost, essentially reduce the problem to (almost) K-independent sequential slot problems. This is far from solving the original slate problem. This is further highlighted by the fact that this method needs to store atmost NK q-values overall.
- Experiments: No comparison against vanilla-SARSA. Given that the reward depends only on the state (viewed item) along with Definition 2, I believe that SARSA would perform better than Vanilla-Q and should be an essential benchmark
Minor typos/Corrections:
- L54: “in the exploitation”
- L151: Missing summation in line 2

---

> ### Author Response · Authors · 2022-07-31
> **Response to reviewer 1QMW**
>
> We thank the reviewer for the critical reading of our manuscript. Since more than one reviewer raised questions about the generality of the decomposition related to the state-dependent cost assumption, we first provide a general answer and then respond to the reviewer's comments in detail. Our method actually solves the general problem without independence assumptions and below are the arguments.
>
> Two important remarks - TO ALL REVIEWERS:
>
> [a] In the rebuttal version we have completely removed the assumption about state dependent cost. This was included in the submitted version as a strong -- but unnecessary -- assumption in our analysis. Besides, a similar assumption was used in [Ie et al, IJCAI'19]. As it turns out, this was a very conservative choice for SlateFree: we have found out that the proofs of Theorem 1 and Theorem 2 do not actually need such an assumption (see rebuttal version, modifications in blue). Instead, we need to introduce in the Theorems the marginal cost-item from Definition 3 $c(s,j)$, and use equality (21) for deterministic policies. (Both the cost-item definition and eq.(21) were already presented in the first submitted version.) In this way we can easily generalize the proofs. The incorporation of the marginal cost-item $c(s,j)$ in the Theorems 1 and 2 slightly modifies the SlateFree-SARSA and -Q methods. Eventually, this explains why, when we use slate-dependent costs in the numerical evaluation section, the SlateFree version converges to the vanilla tabular-Q solution, see last subsection and Fig.2 (right).
>
> [b] Definition 2 is a probability mass function (see Property 3) that results as the marginal of the transition probability given the full slate. This is clearly shown in Property 1, and Property 2, where all the remaining recommendation entries apart from the one occupied by item j are taken in expectation. We do not make any independence assumptions here, we just do a transformation of the original distribution based on marginals.
>
> All these small modifications and comments can be found in the rebuttal version of our work marked with color blue.
>
> ---------------------------------------------
> Specific to reviewer 1QMW
> ---------------------------------------------
>
> - Following the reviewer's suggestion, we have compared our method also against vanilla-SARSA. The plots are in cyan (behind the blue colour of vanilla-Q), and the SARSA code is included in the updated version of the google colab file. We have not observed substantial differences compared to the vanilla-Q method. The text in the rebuttal version is updated accordingly.
>
> - In L151, there is no missing summation; here we use the fact that the probability of the sum of disjoint events (slates) is the probability of the union of these events.
>
> - Indeed the state is the currently (or last) viewed item, so the transition is from the current item to the next one, given the recommended slate. The next state need not belong to the set of recommended items, but can be any item inside the catalog K.
> But, more generally, the state can be the sequence of x (say five) last viewed items. Such an option is possible to be included in our approach, so the state-space would be different than the catalog in this case. We also include a comment about this in the revision.
>
> - The two probabilities mentioned by the reviewer are very different: The probability $P[\omega|s,j]$ is the probability to recommend the specific slate $\omega$, given that we are at state s and in the recommended slate, item j should be included (remember there are more than one slates containing item j). The probability $P[\omega\in A(s;j)|s]$ is the probability that ANY slate $\omega$ that includes item j is recommended, given that we are at state s.
>
> With the above clarifications we hope to have convinced the reviewer that the two Theorems 1 and 2 now have full generality to solve the Slate-MDP. We have updated, as mentioned, the Theorems and RL updates in the rebuttal version to include state and action-slate dependent costs.

---

### Meta-Review · Area_Chair_hZBM · 2022-08-26

**Recommendation:** Reject
**Confidence:** Certain

**Metareview:**

This paper considers reinforcement learning with unordered slate recommendations and shows that this problem can be decomposed into one Q-value per available item as compared to one value per possible slate in existing work. The authors derive a Bellman equation for this formulation and propose model-free algorithms based on it. They show on small synthetic tasks that these methods converge and perform favorably compared to existing methods.

The reviewers appreciated the new decomposition and its potential to enable significantly more efficient algorithms. However, several reviewers also voiced concerns about the clarity of the presentation, the practicality of the approach and the strength of the assumptions. The authors were able to remove a key limiting assumption (costs only depend on state) in the rebuttal revision of the paper. This was viewed very positively and alleviated the concerns about the strong assumptions. However, the concerns about clarity and practicality could not be fully addressed by the authors' response. For this reason, the paper is recommended to be rejected.

Based on the reviewers' comments, the discussions and the AC's own reading of the paper, the following suggestions would make this a very strong paper:

* In the fully general setting where costs are action-dependent, the costs in the Bellman equations are policy dependent and therefore change throughout the execution of the algorithms. As the authors acknowledge, this makes it unclear whether the algorithms provably converge. The authors demonstrate good empirical behavior but their experiments are limited to small toy problems in the absence of function approximation. However, the combination of function approximation and changing (policy-dependent) cost functions may lead to less stable algorithms, a major concern in practice. A theoretical convergence analysis or empirical results with function approximation on more realistic problems would be extremely valuable here.

* The paper lacks a more thorough discussion of the relation to prior works and settings. The questions raised in the reviews around generality and assumptions in this paper shows that readers are left wondering what exactly enables the results as compared to prior work. A better discussion of the exact setting and comparison to other works would be very valuable and a better use of space in the main paper than the short proofs (which could be moved to the appendix).

* The addition of simple illustrative examples would greatly help convey the intuition behind the decomposition.

* The setting in this paper considers the slates being unordered and ordering effects seem to not be captured by this formulation. This is in contrast to existing work. The reader may wonder whether this is crucial for the proposed decomposition. Unordered slates is certainly is a deviation from previous works in this area and most practical recommender systems settings, which would limit the applicability of this approach. Carefully discussing this and if necessary / possible extending this to ordered slates would help and strengthen the paper.

**Award:**

No

---

### Decision · Program_Chairs · 2022-09-14

Reject